# Bax and Bak function as the outer membrane component of the mitochondrial permeability pore in regulating necrotic cell death in mice

Jason Karch[1], Jennifer Q Kwong[1], Adam R Burr[1], Michelle A Sargent[1], John W Elrod[1], Pablo M Peixoto[2], Sonia Martinez-Caballero[2], Hanna Osinska[1], Emily H-Y Cheng[3], Jeffrey Robbins[1], Kathleen W Kinnally[2], Jeffery D Molkentin[1,4]*

[1]Department of Pediatrics, Cincinnati Children's Hospital Medical Center, University of Cincinnati, Cincinnati, United States; [2]Department of Basic Sciences, New York University College of Dentistry, New York, United States; [3]Human Oncology and Pathogenesis Program, Memorial Sloan-Kettering Cancer Center, New York, United States; [4]Howard Hughes Medical Institute, University of Cincinnati, Cincinnati, United States

**Abstract** A critical event in ischemia-based cell death is the opening of the mitochondrial permeability transition pore (MPTP). However, the molecular identity of the components of the MPTP remains unknown. Here, we determined that the Bcl-2 family members Bax and Bak, which are central regulators of apoptotic cell death, are also required for mitochondrial pore-dependent necrotic cell death by facilitating outer membrane permeability of the MPTP. Loss of Bax/Bak reduced outer mitochondrial membrane permeability and conductance without altering inner membrane MPTP function, resulting in resistance to mitochondrial calcium overload and necrotic cell death. Reconstitution with mutants of Bax that cannot oligomerize and form apoptotic pores, but still enhance outer membrane permeability, permitted MPTP-dependent mitochondrial swelling and restored necrotic cell death. Our data predict that the MPTP is an inner membrane regulated process, although in the absence of Bax/Bak the outer membrane resists swelling and prevents organelle rupture to prevent cell death.

*For correspondence: jeff.
molkentin@cchmc.org

Competing interests: The authors declare that no competing interests exist.

## Introduction

Mitochondria have emerged as critical regulators of apoptotic and necrotic cell death, with distinct molecular effectors underlying each process (*Tait and Green, 2010*). Apoptosis is both an ATP- and caspase-dependent event characterized by chromatin condensation, cell shrinkage, and plasma membrane blebbing (*Danial and Korsmeyer, 2004*). Necrosis, however, is an ATP-independent event characterized by organelle swelling and early plasma membrane rupture (*Danial and Korsmeyer, 2004*). One key difference between apoptosis and necrosis is the type of events that occur in the mitochondria and the distinct pores that are formed. During apoptosis, the mitochondrial outer membrane is permeabilized through the action of the Bcl-2 family members Bax/Bak that generate large pores that allow for the release of cytochrome *c* and the subsequent activation of the caspase cascade. In contrast, during forms of cellular necrosis, one key regulated event is the opening of the mitochondrial permeability transition pore (MPTP), a protein complex that was proposed to span the inner and outer mitochondrial membranes in facilitating loss of the inner membrane potential, swelling, and eventual rupture of the organelle (*Halestrap, 2009*; *Tait and Green, 2010*). While the identity of the outer and

**eLife digest** In all multicellular plants and animals, cells are continuously dying and being replaced. There are a number of different types of cell death, but two of the best studied are apoptosis and necrosis. Apoptosis, sometimes referred to as 'cell suicide', is a form of programmed cell death that is generally beneficial to the organism. Necrosis, however, occurs whenever cells are damaged—for example, due to a lack of oxygen—and can trigger harmful inflammation in surrounding tissue. Although the processes leading up to apoptosis and necrosis are very different, they both involve regulated changes in mitochondria—the organelles that supply cells with chemical energy.

Mitochondria have a distinctive appearance, being enclosed by two membranes, the innermost of which is highly folded. During apoptosis, large pores form in the outer membranes of mitochondria. These pores are generated by two proteins—Bax and Bak—and they enable the mitochondrion to release proteins that activate processes involved in apoptosis. Pores also form in the mitochondrial membrane during necrosis. However, these mitochondrial permeability transition pores (MPTPs) occur simultaneously in both the inner and outer membranes and are thought to lead to swelling and rupture of mitochondria.

Now, Karch et al. have shown that Bax and Bak are also involved in the formation of these permeability pores that underlie necrosis. When mouse cells that had been genetically modified to lack Bak and Bax were grown in cell culture, they were found to be resistant to substances that normally induce necrosis. Instead, their mitochondria continued to function normally, suggesting that MPTPs cannot form in the absence of Bak and Bax.

Karch et al. then generated mice with heart cells that lack Bax and Bak, and deprived their hearts of oxygen to simulate a heart attack. Compared to normal mice, the genetically modified animals experienced less damage to their heart muscle, suggesting that the absence of Bax and Bak prevents cell death due to necrosis. If Bax and Bak are involved in both apoptosis and necrosis, inhibiting them could be a powerful therapeutic approach for preventing all forms of cell death during heart attacks or in certain degenerative diseases.

inner membrane components of the MPTP remains elusive, cyclophilin D (CypD), a peptidylprolyl isomerase located in the mitochondrial matrix, is known to bind and regulate the inner membrane complex. Indeed, deletion of the gene encoding CypD renders mitochondria resistant to $Ca^{2+}$ overload–induced swelling and the heart and brain partially resistant to cell death due to ischemic injury (*Baines et al., 2005*; *Nakagawa et al., 2005*; *Schinzel et al., 2005*).

Apoptosis at the level of the mitochondria is absolutely dependent on Bax/Bak, as deficiency in the genes encoding these two proteins renders cells resistant to apoptosis and concomitant release of cyto-chrome *c* through the outer mitochondrial membrane (*Tait and Green, 2010*). Previous studies have suggested a role for Bax/Bak in the regulation or formation of the MPTP (*Marzo et al., 1998*; *Narita et al., 1998*), although more recent studies have disputed such a direct role, suggesting that Bax/Bak only have a secondary effect, such as by regulating mitochondrial fission/fusion or no effect whatsoever, or just a secondary effect due to organelle rupture (*De Marchi et al., 2004*; *Vaseva et al., 2012*; *Whelan et al., 2012*). Here, we show that Bax/Bak are required for mitochondrial permeability pore-dependent cell death by serving as a necessary functional component of the MPTP within the outer mitochondrial membrane in a manner that is distinct from their more active mechanism of oligomerization during apoptosis, placing Bax/Bak at the bifurcation point of mitochondrial-dependent apoptosis and necrosis.

## Results

### Bax/Bak1 null cells resist necrotic cell death

We first evaluated the ability of wild type (Wt) and *Bax/Bak1*-deficient (*Bak1* gene encodes Bak protein) mouse embryonic fibroblasts (MEFs) to die by apoptosis or mitochondrial-dependent necrosis. As previously reported, *Bax/Bak1* double-knockout (DKO) MEFs were highly resistant to cell death by the apoptotic inducer staurosporine (*Wei et al., 2001*), while Wt MEFs showed high levels of killing (*Figure 1A*). However, DKO MEFs were also resistant to $H_2O_2$, ionomycin, and DNA alkylation

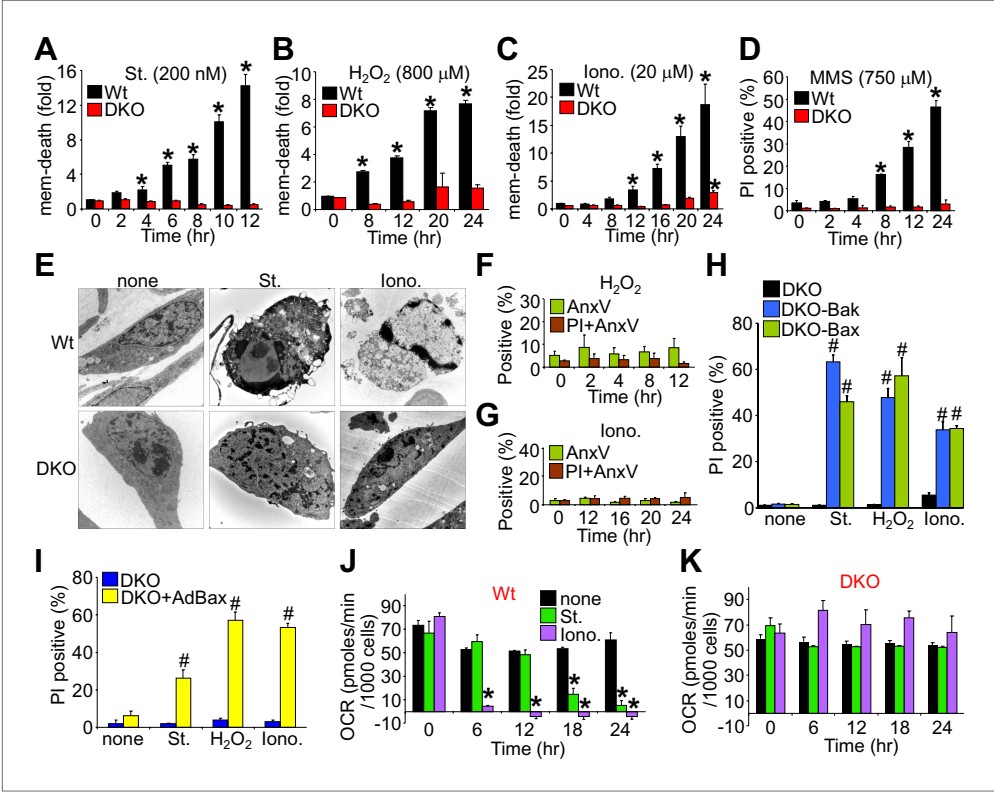

**Figure 1**. *Bax/Bak1* DKO MEFs are resistant to a necrotic-like cell death. (**A**)–(**C**) MultiTox-Fluor multiplex cytotoxicity assay, which measures membrane (mem) integrity loss induced death, for different time points in cultures of Wt and *Bax/Bak1* DKO MEFs treated with staurosporine (St), $H_2O_2$, ionomycin (Iono) for different time points. (**D**) Propidium iodide (PI) inclusion to assess membrane integrity following methyl methanesulfonate (MMS) for different time points. (**E**) Transmission electron microscopy of Wt and DKO MEFs treated with the indicated agents for 10 hr (St, 200 nM) or 20 hr (Iono, 20 μM). Magnification is ×10,000 for all panels. (**F**) and (**G**) FACS quantitation of annexin V and PI staining of DKO MEFs treated with $H_2O_2$ (800 μM) or ionomycin (20 μM) for the indicated time points. Apoptotic and necrotic killing was almost nonexistent in DKO MEFs compared with Wt MEFs shown in *Figure 1—figure supplement 1*. (**H**) PI inclusion rates for cell death assessment in DKO MEFs or DKO MEFs expressing a stable cDNA for Bax or Bak with the indicated death inducing agents for 20 hr. #p<0.05 vs DKO alone. (**I**) PI inclusion rates for cell death assessment in DKO MEFs and DKO MEFs infected with a Bax expression adenovirus following stimulation with the indicated death-inducing agents for 12 (St) or 24 hr ($H_2O_2$, Iono). #p<0.05 vs DKO alone. (**J**) and (**K**) Oxygen consumption rates (OCR) in cultures of Wt (**J**) or DKO (**K**) MEFs treated with St (200 nM) or Iono (20 μM) for the indicated time points. Rates are expressed as pmol/min per 1000 cells in a well of a 24-well dish. All assays were performed in duplicate and averaged from three independent experiments. *p<0.05 vs 0 time point; #p<0.05 vs no treatment.

The following figure supplements are available for figure 1:

**Figure supplement 1**. Conditions whereby staurosporine induces only apoptosis while ionomycin induces necrosis in cultured Wt MEFs.

**Figure supplement 2**. *Bax/Bax1* null MEFs (DKO) have normal mitochondrial protein content except for a slight upregulation of Bim and BNip3.

**Figure supplement 3**. *Bax/Bak1*-deficient mitochondria remain coupled for ATP generation after ionomycin treatment.

with methyl methanesulfonate (MMS) that appeared to induce a necrotic phenotype (*Figure 1B–D*). Indeed, electron microscopy verified that staurosporine-treated Wt cells, but not DKO cells, exhibited hallmarks of apoptosis including nuclear condensation with maintenance of plasma membrane integrity, while ionomycin induced a purely necrotic phenotype that showed rupture of the plasma membrane and dispersion of the nucleus, which was not seen in DKO cells (*Figure 1E*). By comparison, analysis

of apoptotic cell death with annexin V (phosphatidylserine externalization in the plasma membrane) and necrotic death with propidium iodide (PI, plasma membrane opening/rupture) staining in cultured MEFs showed an apoptotic profile with staurosporine treatment, while ionomycin induced a necrotic phenotype; $H_2O_2$ had an intermediate profile of both forms (**Figure 1—figure supplement 1A–D**). Apoptosis leads to annexin V staining prior to PI staining, while necrosis causes simultaneous annexin V and PI staining. Molecular markers also confirmed a uniform apoptotic profile in Wt MEFs treated with staurosporine, such that caspase cleavage was observed and cell death was inhibited with z-Vad (pan-caspase inhibitor) but not by disabling the MPTP in MEFs from *Ppif* (CypD)-null mice (**Figure 1—figure supplement 1E–H**). By comparison, ionomycin-induced cell death in Wt MEFs did not appreciably involve caspase activation nor was it affected by caspase inhibition, while inhibiting MPTP function by deletion of the *Ppif* gene–reduced cell death, suggesting mitochondrial-dependent necrosis (**Figure 1—figure supplement 1E–H**). Using these conditions, DKO MEFs were completely refractory to $H_2O_2$- and ionomycin-induced necrosis as assessed with annexin V and PI labeling (**Figure 1F,G**; compare with **Figure 1—figure supplement 1C,D**), but all deaths were restored in DKO MEFs containing a stably integrated Bax or Bak viral–based expression cassette, or acutely by infection with a recombinant Bax encoding adenovirus, collectively indicating that there is not an unrelated defect in the DKO MEFs (**Figure 1H,I**). Reconstitution with Bax and Bak produced an equal or slightly lower level of protein compared with endogenous protein content in Wt cells (data not shown).

Apoptotic cell death proceeds with ATP generation and cellular respiration, as the inner mitochondrial membrane typically remains intact for a period of time after the outer membrane is permeabilized. Necrosis more immediately extinguishes oxidative phosphorylation as the inner mitochondrial membrane is opened and the proton gradient is lost. Indeed, Wt MEFs treated with staurosporine maintained respiration considerably longer than observed with ionomycin treatment even though staurosporine-treated Wt MEFs died considerably faster than those with ionomycin (**Figure 1J**). However, DKO MEFs showed no reduction in oxygen consumption with ionomycin over 24 hr, providing yet another line of evidence that MPTP-dependent mitochondrial dysfunction does not occur in the absence of Bax/Bak protein (**Figure 1K**). Other mitochondrial proteins involved in cell death, the MPTP, and respiration all appeared normal in DKO MEFs, except for a mild upregulation of Bim and Bnip3 (**Figure 1—figure supplement 2**). We also verified that ionomycin treated *Bax/Bak1* null MEFs still had functionally competent mitochondria as compared to ionomycin treated Wt MEFs by measuring respiration in response to oligomycin, FCCP, and antimycin A. Both oligomycin (complex V inhibitor) and antimycin A (complex III inhibitor) inhibited oxygen consumption, suggesting that respiration observed was due to mitochondrial oxygen consumption as opposed to extra-mitochondrial sources. Further, treatment with FCCP increased respiration in the *Bax/Bak1* null but not in the Wt MEFs, suggesting that mito-chondrial respiration was still coupled to ATP synthesis in ionomycin treated DKO cells (**Figure 1—figure supplement 3**). Thus, mitochondria from *Bax/Bak1* null MEFs were fully functional after ionomycin and responded to the drugs similar to untreated mitochondria, showing their ability to maintain respiration and generate ATP long after Wt mitochondria have lost such ability with ionomycin treatment (**Figure 1—figure supplement 3**). Finally, and consistent with results in DKO MEFs, cardiac-specific deletion of *Bax/Bak1* significantly protected the heart from ischemia–reperfusion (I-R) injury and reduced lethality in mice subjected to permanent myocardial infarction injury (**Figure 2—figure supplement 1**). These results indicate that loss of Bax/Bak protein protects from ischemia-induced necrotic cell death in vivo due to a disease relevant stimuli.

## Mitochondria lacking Bax/Bak are resistant to MPTP-dependent swelling

In response to ischemia and $Ca^{2+}$ overload, CypD-regulated opening of the MPTP leads to mitochondrial swelling (**Halestrap, 2009**). Examination of this effect in purified mitochondria isolated from Wt MEFs showed swelling with acute $Ca^{2+}$ or atractyloside administration, which was blocked with cyclosporine (CsA), a CypD inhibitor (**Figure 2A,C**). However, purified mitochondria from DKO MEFs failed to show $Ca^{2+}$ or atractyloside-induced swelling under the same conditions (**Figure 2B,D**). This difference in $Ca^{2+}$-induced swelling is even more pronounced in mitochondria freshly isolated from Wt or single *Bak1*$^{-/-}$ mouse hearts or livers, which showed abundant swelling (**Figure 2E,F,H,I**). However, mitochondria purified from mouse hearts or livers of *Bak1/Bax-loxP* targeted mice subjected to heart- or liver-specific deletion with the appropriate Cre-expressing transgenes (**Figure 2—figure supplement 1 and 2**) were resistant to $Ca^{2+}$-induced swelling (**Figure 2G,J**). Consistent with these results, purified DKO MEF

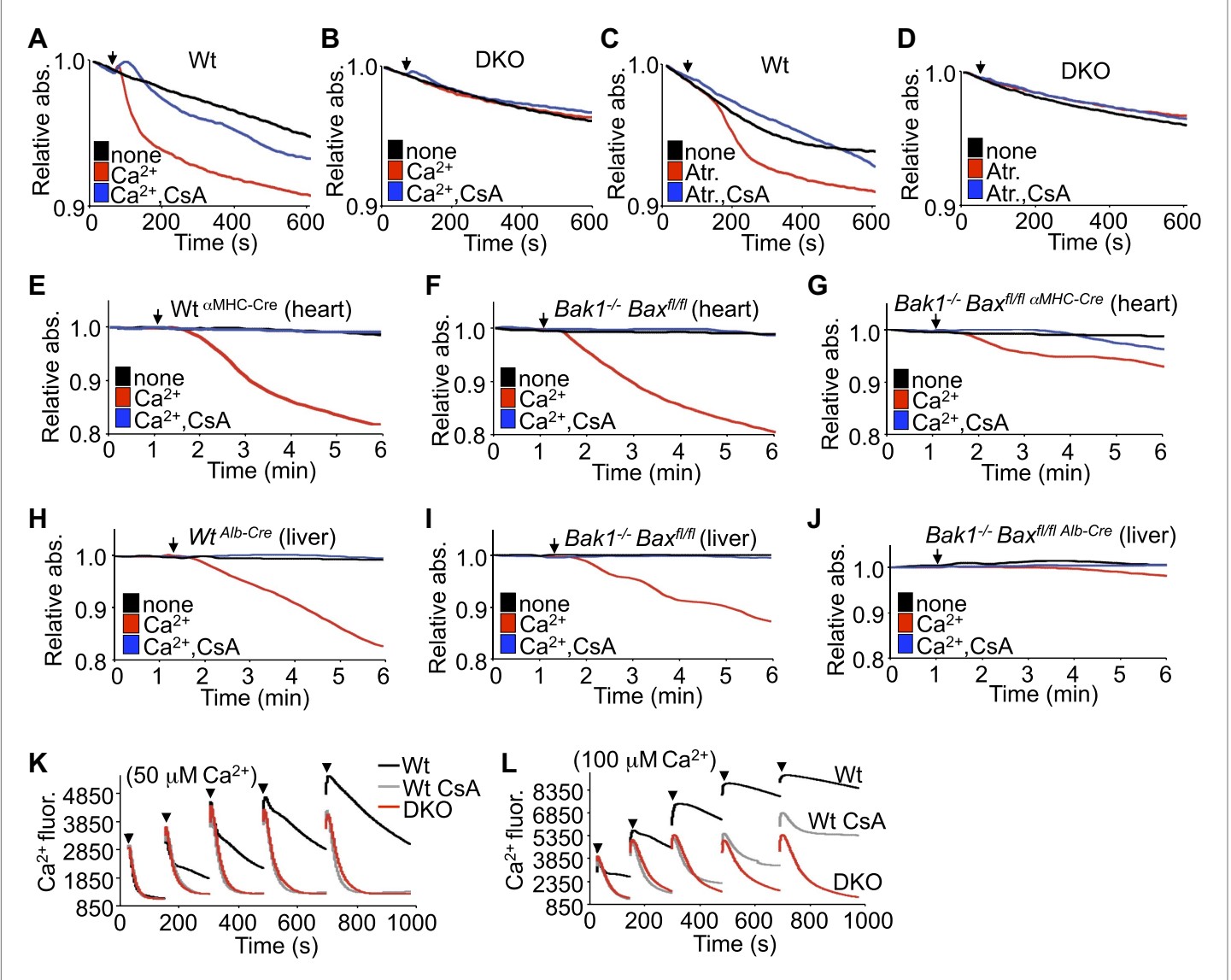

**Figure 2**. Bax/Bak-deficient mitochondria are resistant to swelling and MPTP formation. (**A**)–(**D**) Mitochondrial absorbance change swelling assay upon Ca$^{2+}$ (**A** and **B**) or atractyloside (Atr. **C** and **D**) addition (arrowhead) in mitochondria purified from Wt or DKO MEFs. CsA (2 µM) is given to desensitize MPTP-induced swelling as a control. (**E**)–(**G**) Swelling assays in mitochondria purified from mouse hearts of the indicated genotypes. Ca$^{2+}$ is added at a given time (arrowhead), and CsA (2 µM) is a control that desensitizes MPTP formation and swelling. (**H**)–(**J**) Swelling assays in mitochondria purified from mouse livers of the indicated genotypes. Ca$^{2+}$ is added at a given time (arrowhead), and CsA (2 µM) is a control that desensitizes MPTP formation and swelling. (**K**) and (**L**) Ca$^{2+}$ uptake capacity assay with the external Ca$^{2+}$ indicator dye calcium green-5N with purified mitochondria from Wt or DKO MEFs. 50 (**K**) or 100 µM (**L**) cumulative Ca$^{2+}$ additions are shown at each arrowhead. Fluorescence diminishes as the mitochondria remove the Ca$^{2+}$ from the solution until the MPTP opens and Ca$^{2+}$ is no longer sequestered. CsA is given to Wt mitochondria as a control to show the closed state of the MPTP. The swelling and Ca$^{2+}$ uptake assays were performed in three independent experiments, although representative tracings are shown.

The following figure supplements are available for figure 2:

**Figure supplement 1**. Cardiac-specific *Bax/Bak1* deficiency renders the heart partially resistant to cell death after ischemic reperfusion (I-R) injury.

**Figure supplement 2**. Liver-specific *Bax/Bak1* deletion with the albumin-Cre transgene.

**Figure supplement 3**. Atomic absorption mass spectrometry from purified mitochondria from Wt or *Bak1$^{−/−}$ Bax$^{Alb-Cre}$* livers.

**Figure supplement 4**. Bax/Bak-deficient mitochondria are resistant to Ca$^{2+}$-induced swelling but are otherwise capable of nonspecific swelling.

mitochondria continue to take-up exogenous $Ca^{2+}$ while mitochondria isolated from Wt MEFs show gradual inhibition of $Ca^{2+}$ uptake at the 50- and 100-μM pulses, such that the MPTP opens and the mitochondria achieve equilibrium with the test solution at less cumulative $Ca^{2+}$ levels (**Figure 2K,L**). Direct inhibition of the MPTP with CsA enhanced Wt mitochondrial $Ca^{2+}$ uptake because the MPTP remains closed through a higher range of $Ca^{2+}$, and by comparison, *Bax/Bak1* deficiency was also highly potent in allowing progressive $Ca^{2+}$ uptake without engaging the MPTP (**Figure 2L**). Indeed, mass spectrometry analysis of $Ca^{2+}$ and other cations showed that basal $Ca^{2+}$ was enhanced in Bax/Bak-deficient mitochondria compared with Wt, and that with $Ca^{2+}$ addition, Bax/Bak-deficient mitochondria maintained more of this ion beyond any additional effect with CsA (**Figure 2—figure supplement 3**). Since MPTP formation has been proposed to function as a $Ca^{2+}$ release mechanism (**Elrod et al., 2010**; **Bernardi and von Stockum, 2012**), the observed augmentation in baseline $Ca^{2+}$ in Bax/Bak-deficient mitochondria, and their greater uptake capacity further suggest that Bax/Bak are required for MPTP-dependent $Ca^{2+}$ release.

We instituted additional control experiments to further verify that mitochondria from either *Bax/Bak1* null MEFs or deficient livers were otherwise uncompromised. Alamethicin, which nonspecifically permeabilizes mitochondria, showed equal swelling between mitochondria from both DKO and Wt MEFs or livers (**Figure 2—figure supplement 4A,B,E,F**). These results indicate that mitochondria lacking Bax/Bak are still capable of swelling if uniformly permeabilized in a non-MPTP-dependent process. However, in these same experiments, DKO mitochondria were still highly resistant to $Ca^{2+}$-induced swelling, whether viewed as the raw tracings or when normalized. Finally, different concentrations of KCl buffer were also used to generate a range of mitochondrial osmotic states, which showed no difference in absorbance between mitochondria isolated from Wt and DKO MEFs or livers, hence loss of Bax/Bak did not adversely affect mitochondrial swelling in this manner, suggesting that despite being insensitive to $Ca^{2+}$-induced MPTP opening, they are otherwise normal (**Figure 2—figure supplement 4C,D,G,H**).

## Bax/Bak regulate cell death independent of oligomerization

Upon apoptotic stimulation, Bax/Bak become activated and oligomerize in the outer mitochondrial membrane to generate large pores that release cytochrome *c* (**Tait and Green, 2010**). Here, we investigated Bax or Bak activation and oligomerization using our conditions for apoptotic (staurosporine) or necrotic (ionomycin) killing. Staurosporine caused the known conformational shift in both Bax and Bak in their N-termini that facilitates activation, but ionomycin caused no such effect (**Figure 3A,B**). Consistent with these results, staurosporine but not ionomycin caused oligomerization of Bax into larger complexes assessed by gel filtration (**Figure 3C**).

Our results suggest that the presence of Bax or Bak in the outer membrane, even if inactive as defined for apoptotic stimuli, is permissive for mitochondrial swelling by facilitating permeability. Indeed, Bcl-2 family members are known to alter membrane characteristics and have channel-like activities of multiple conductance states (**Antonsson et al., 1997**; **Minn et al., 1997**; **Schlesinger et al., 1997**). Here, we examined the ability of two nonoligomerizing mutants of Bax (**George et al., 2007**) to mediate cell death in the DKO MEF background. As a control, DKO MEFs reconstituted with a retrovirus expressing Wt Bax showed restoration of both staurosporine-mediated (apoptotic) and ionomycin-mediated (necrotic) killing (**Figure 3D**). However, restoration with either of the two mutants of Bax that cannot oligomerize, although both still localize to the mitochondria (**George et al., 2007**; **Hoppins et al., 2011**), restored ionomycin-induced killing but not apoptotic killing with staurosporine (**Figure 3D**). More importantly, purified mitochondria from these DKO MEFs reconstituted with Wt or the nonoligomerizing Bax mutants (**Hoppins et al., 2011**) each showed partial restoration of MPTP function in the $Ca^{2+}$ uptake assay (**Figure 3E**). Specifically, purified mitochondria from DKO MEFs pulsed with 75 μM $Ca^{2+}$ continued to show progressive $Ca^{2+}$ uptake, while mitochondria from Wt or the two nonoligomerizing Bax mutants each showed early saturation with $Ca^{2+}$ (**Figure 3E**). These results indicate that Bax can reconstitute MPTP function and permit nonapoptotic death in a nonoligomerized conformation, likely by affecting the permeability characteristics of the outer mitochondrial membrane.

## Bax/Bak permit permeability of outer mitochondrial membrane

To examine the structural basis for how Bax/Bak might control MPTP-dependent cell death through the outer mitochondrial membrane, we first performed electron microscopy on isolated mitochondria. Purified Wt liver mitochondria treated with $Ca^{2+}$ show a characteristic swelling profile and a loss of

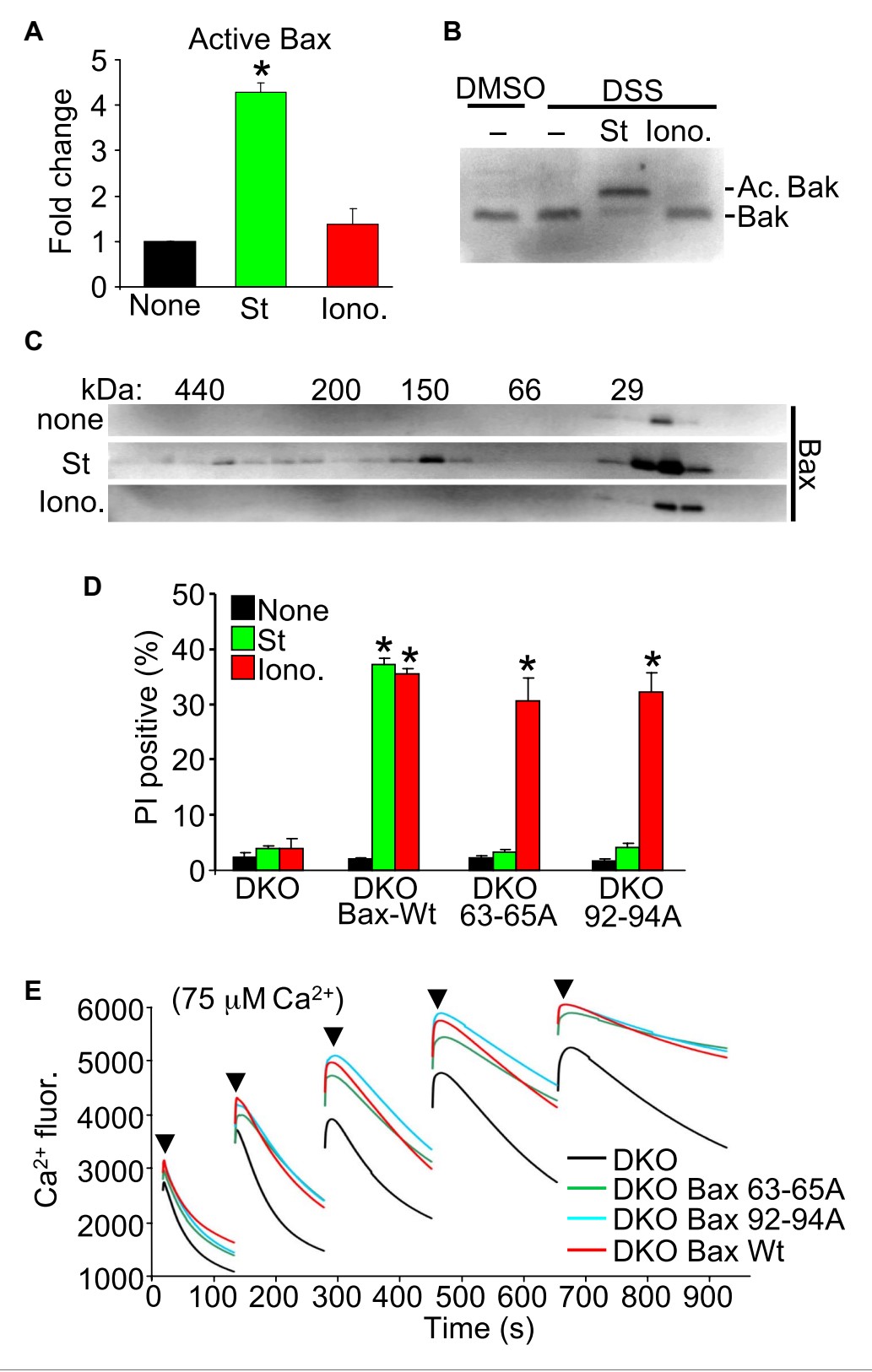

**Figure 3**. Nonoligomerized Bax/Bak mediate mitochondrial swelling and necrosis. (**A**) Quantitation by FACS analysis of Wt MEFs treated previously with the indicated stimuli. Sorting was with the activated Bax epitope mAB 6A7. The results were averaged from three independent experiments. (**B**) Western blotting for activated Bak from *Figure 3. Continued on next page*

*Figure 3. Continued*

fixed cells (DSS crosslinker) that were previously stimulated with staurosporine (200 nM for 10 hr) or ionomycin (20 µM for 20 hr). (**C**) Western blotting for Bax after gel filtration chromatography to show increasing molecular weights of complexes in cells stimulated previously with staurosporine (200 nM for 10 hr) or ionomycin (20 µM for 20 hr). One of three independent experiments is shown, all with similar results. (**D**) PI incorporation cell death assay in DKO MEFs at baseline (control), DKO MEFs reconstituted with Wt Bax, or DKO MEFs reconstituted with Bax mutants (amino acids 63–65 or 92–94 were mutated to alanines) that cannot oligomerize and generate apoptotic pores in the outer mitochondrial membrane. MEFs were stimulated with staurosporine (200 nM for 24 hr) or ionomycin (20 µM for 24 hr). The results were averaged from three independent experiments. (**E**) Fluorescence reading of $Ca^{2+}$ measured with calcium green-5N indicator in solution in the presence of purified mitochondria from the indicated MEFs. Cumulative $Ca^{2+}$ additions are shown at each arrowhead. The assay was performed in three independent experiments, although representative tracings are shown. *$p<0.05$ vs none.

inner membrane cristae, which was prevented by inhibition of the MPTP with CsA (*Figure 4A,B*). Purified *Bax/Bak1*-deficient liver mitochondria appeared morphologically normal at baseline, although in response to a $Ca^{2+}$ challenge, they did not swell, despite inner membrane cristae reorganization that was still inhibited with CsA (*Figure 4A,B*). This observation suggests that in the absence of Bax/Bak, the inner membrane still undergoes MPTP-associated cristae reorganization, but mitochondrial swelling and rupture are prevented.

We more directly examined the manner in which the outer membrane could affect the inner membrane and MPTP function. Specifically, in the absence of Bax/Bak protein, mitochondrial inner membrane opening still occurred in response to ionomycin as measured directly with the calcein–$CoCl_2$ assay, similar to Wt MEFs (*Figure 4—figure supplement 1A*). This later observation is consistent with ultrastructical data discussed above in which the inner membrane can still reorganize without Bax/Bak in response to necrotic stimuli. Notably, purified mitochondria from Wt MEFs showed release of cytochrome *c* with ionomycin stimulation and loss of respiration with serial addition of $Ca^{2+}$, while mitochondria lacking Bax and Bak protein did not show cytochrome *c* release or loss of respiration (*Figure 4—figure supplement 1B,C*). These results suggest that the inner membrane component of the MPTP still functions normally in *Bax/Bak1* DKO MEFs, but without sufficient outer membrane permeability, swelling/rupture is prevented and the mitochondria can reestablish respiration capacity.

To more carefully examine this concept, we attempted to reconstitute the outer membrane function of Bax/Bak with other pore-forming entities or nonoligomerized recombinant Bax (rBax). Purified liver DKO mitochondria stimulated with $Ca^{2+}$ were still protected from swelling (*Figure 4C,D*). However, treatment of DKO mitochondria with rBax at concentrations that did not cause cytochrome *c* release (data not shown) fully restored $Ca^{2+}$-induced mitochondrial swelling, which was still inhibited with CsA (*Figure 4C,D*). Moreover, treatment with gossypol, a BH3-mimetic compound that can convert Bcl-2 and Bcl-xl into pore-forming pro-cell death agents (*Lei et al., 2006*) restored mitochondrial swelling in the absence of Bax/Bak protein, which was also inhibited with CsA (*Figure 4C,D*). Generation of nonspecific pores that only affected the outer mitochondrial membrane with either digitonin or tetanolysin also restored $Ca^{2+}$-induced swelling in DKO mitochondria, which was again inhibited with CsA (*Figure 4C,D*). These results suggest that Bax/Bak permit mitochondrial swelling through a permeabilization activity in the outer mitochondrial membrane that functions independently of the inner membrane CypD-regulated component of the MPTP. Indeed, increasing concentrations of poloxamer-188 (P-188), which inherently reduces membrane permeability characteristics (*Wu et al., 2004*), prevented mitochondrial swelling induced by $Ca^{2+}$ as analyzed in purified mitochondria in solution or by transmission electron microscopy (*Figure 4E,F*). These observations suggest that outer membrane permeability plays a necessary and permissive role in MPTP-dependent swelling, rupture, and cellular necrosis.

We also performed patch clamping of the outer mitochondrial membrane from Wt and DKO purified mitochondria to directly examine if Bax/Bak were generating a pore-like activity in the outer membrane. The MPTP has been well characterized by direct patch-clamping of mitochondria and shown to have a specific conductance profile that is distinct from any other channels or permeating activities (*Kinnally and Antonsson, 2007*). The data show baseline conductance of approximately 750 pS in Wt mitochondria, which was significantly reduced in DKO mitochondria to approximately 300 pS (*Figure 5A,C*). Addition of the same non-oligomerizing rBax restored this baseline permeability and

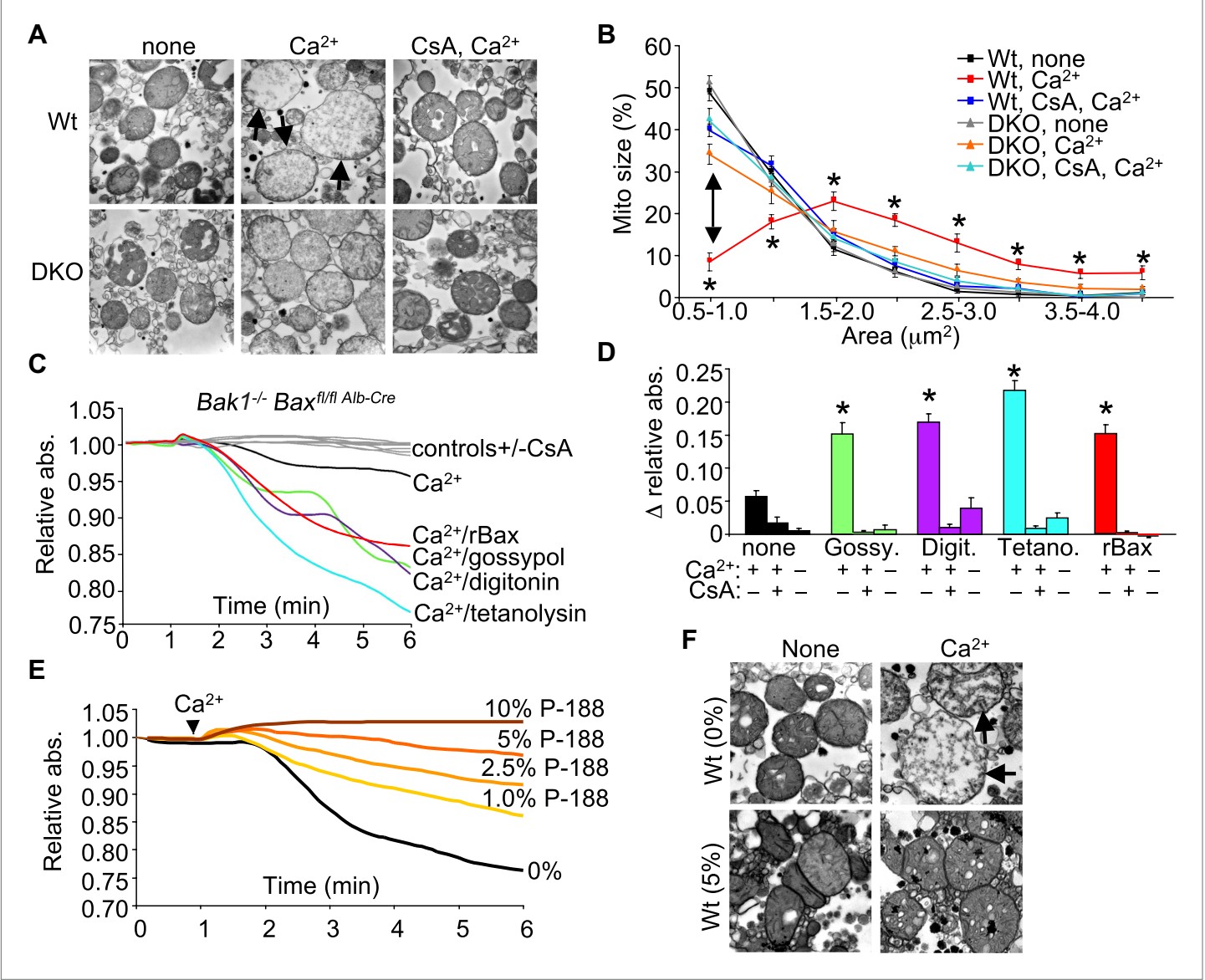

**Figure 4**. Bax/Bak-deficient mitochondria are defective in outer membrane permeability and associated swelling. (**A**) Transmission electron microscopy (EM) of purified Wt and *Bax/Bak1*-deficient (DKO) liver mitochondria at baseline (none) or with $Ca^{2+}$ with or without CsA (2 μM) for 5 min prior to fixation. The arrows show swollen and rupturing mitochondria. Magnification is ×40,000 for all panels. (**B**) Quantitation of mitochondrial cross-sectional area in different quartiles from the type of EM data shown in (**A**). (**C**) Absorbance reading for swelling in liver-derived DKO mitochondria treated with the indicated conditions. The gray lines are the controls that represent all five $Ca^{2+}$ stimulated conditions with CsA, or DKO mitochondria not stimulated with $Ca^{2+}$. (**D**) Quantitation of the change in absorbance tracings shown in (**C**) for mitochondrial swelling under the indicated conditions. Four independent swelling experiments were tabulated. *$p<0.05$ vs DKO + $Ca^{2+}$ only. (**E**) Absorbance reading for swelling in liver-derived Wt mitochondria treated with $Ca^{2+}$ and increasing concentration of poloxamer 188 (P-188). (**F**) EM of purified Wt liver mitochondria with and without $Ca^{2+}$, with and without 5% P-188. The arrows show swollen and rupturing mitochondria. Magnification is ×40,000 for all panels.

The following figure supplements are available for figure 4:

**Figure supplement 1**. *Bax/Bak1* DKO MEF are fully susceptible to inner membrane permeability, but resist full MPTP with $Ca^{2+}$ overload.

even enhanced it slightly (***Figure 5A,C***). This low-conductance channel activity with rBax was slightly cation selective but was not voltage dependent (±50 mV) (data not shown). These results suggest that the presence of Bax/Bak impart a different permeability characteristic to the outer mitochondrial membrane that relates to its necessary but permissive role in MPTP function. Because Bax/Bak could

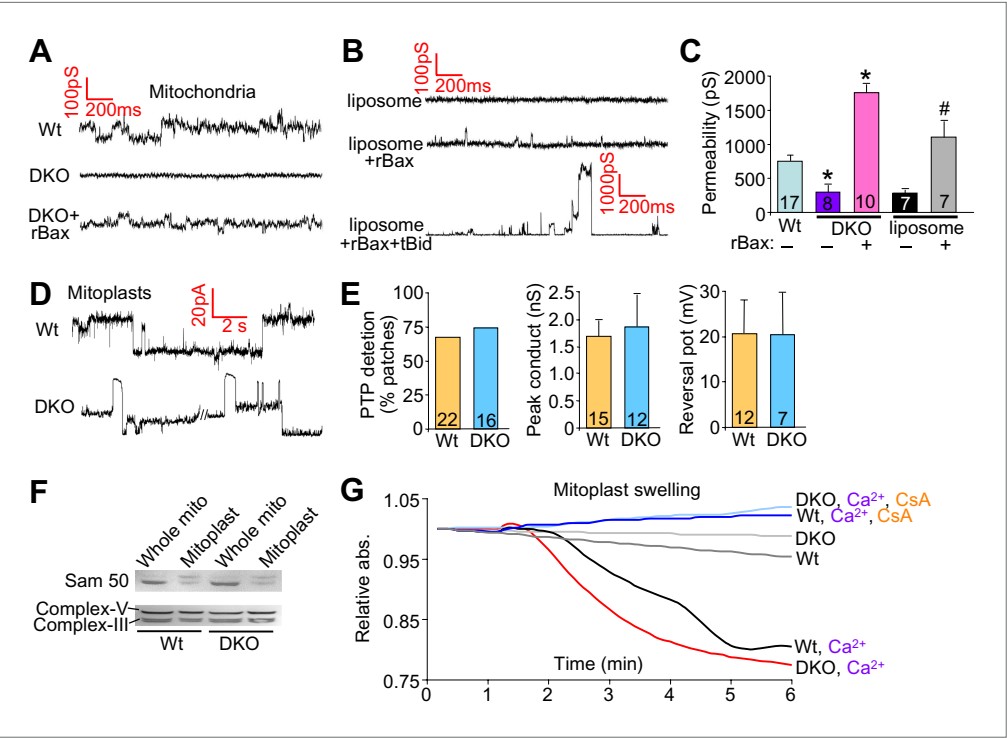

**Figure 5**. Bax/Bak regulate outer mitochondrial membrane permeability directly. (**A**) and (**B**) Patch clamp current traces (2 s, 5 KHz sampling, 1 KHz filter in 0.15 M KCl media) from Wt and DKO whole mitochondria and liposomes are shown after excised patches were voltage clamped at −20 to 40 mV and reveal small variable-sized transitions, typically approximately 50–100 pS. Recombinant Bax (rBax at 50–100 ng/μl) was included in the patch pipette as indicated. Current trace recorded from a liposome with a patch pipette backfilled with media containing 10 nM rBax + 10 nM tBid reveals up to 3000 pS transitions, which are not seen with rBax alone. (**C**) Graph of multiple independent patch recordings of the representative traces shown in (**A** and **B**). Large channel activities, such as from TOM, were excluded from the Wt and DKO mitochondrial data set. *p<0.05 vs Wt mitochondria; #p<0.05 vs untreated liposomes. (**D**) Patch clamp current traces from mitoplasts (inner membrane) were recorded as in (**A**) from Wt and DKO mitochondria at −60 and −20 mV, respectively. (**E**) Graphs showing comparisons of MPTP single-channel characteristics recorded from the indicated number of independent patches of mitoplasts isolated from Wt and DKO cells. MPTP was scored present from mitoplast recordings if the peak conductance was ≥1 nS, transition sizes ≥0.3 nS, voltage dependence, and when possible, cation selectivity. (**F**) Western blot for the outer and inner mitochondrial membrane proteins Sam-50 and complex V/III, respectively. Samples were normalized for complex V/III to show differences in Sam-50 and proper enrichment of inner membrane in the mitoplast preparation vs whole mitochondria. (**G**) Absorbance reading for swelling in liver-derived Wt and DKO mitoplasts (outer membranes removed) treated with the indicated conditions.

be acting through another protein in the outer mitochondrial membrane to impart this increase in permeability, we also used a reconstitution assay in liposomes (*Figure 5B,C*). Remarkably, addition of monomeric rBax significantly increased the basal permeability of liposomes, but to a level, that was still more than 10× less than when rBax was forced to oligomerize with recombinant tBid treatment (*Figure 5B,C*). The high resistance and low permeability of the outer mitochondrial membrane with Bax/Bak deficiency reported here and previously (*Martinez-Caballero et al., 2009*) suggest that this membrane is not a sieve but rather a membrane whose permeability is tightly regulated, despite the presence of other channels such as voltage-dependent anion channel (VDAC, comprising as many as three gene products, *Vdac1*, *Vdac2*, and *Vdac3*) and TOM (*Jonas et al., 2004*; *Martinez-Caballero et al., 2005*).

Importantly, direct patching of the inner mitochondrial membrane in mitoplasts (outer mitochondrial membranes are removed) showed that Bax/Bak do not directly affect the MPTP at the level of the inner membrane (*Figure 5D,E*). Indeed, the MPTP from the inner membrane of Wt mitochondria showed no difference in frequency, conductance peak, or reversal potential with Bax/Bak null mitochondria (*Figure 5D,E*). Direct analysis of Ca$^{2+}$-induced swelling of purified mitoplasts (outer membranes removed) showed that both Wt and DKO preparations swelled to the same extent in a manner that

was still inhibited with CsA (*Figure 5G*). Western blotting confirmed the integrity of the mitoplast preparation (*Figure 5F*). These final observations further indicate that Bax and Bak do not directly regulate the inner membrane aspects of the MPTP, further supporting their more dedicated function within the outer mitochondrial membrane in permitting mitochondrial swelling and end-stage MPTP that leads to organelle rupture and cell death.

## Bax/Bak regulate cell death independent of ER Ca$^{2+}$

*Bax/Bak1* DKO cells were previously shown to have reduced Ca$^{2+}$ levels in the endoplasmic reticulum (ER) as a protective mechanism against cell death (*Scorrano et al., 2003*). We also examined this effect to determine if it might secondarily influence necrotic cell death in the DKO MEFs, or in purified mitochondria from these cells. Total intracellular Ca$^{2+}$ levels were similar in DKO MEFs compared with Wt MEFs, as was ER Ca$^{2+}$ levels measured with two different agonists (thapsigargin or ATP), although total mitochondrial Ca$^{2+}$ levels were significantly elevated in DKO MEFs (*Figure 6A–D*). It is uncertain why we failed to observe a decrease in ER Ca$^{2+}$ in *Bak/Bak1* DKO MEFs, although our conditions were different from those previously reported, as we measured individual living cells in adherent cultures and used fivefold higher levels of thapsigargin. To unequivocally rule out a secondary effect due to lower levels of ER Ca$^{2+}$ in DKO MEFs, we also used a recombinant adenovirus expressing the sarcoplasmic reticulum Ca$^{2+}$ ATPase 1 (SERCA1) to load the ER with even more Ca$^{2+}$, as previously shown (*Scorrano et al., 2003*). However, AdSERCA1 infection neither restored cell death with ionomycin or staurosporine administration in DKO MEFs nor did it increase cell death in Wt MEFs, although it did load cells with significantly more Ca$^{2+}$ (*Figure 6E* and data not shown). Hence, we do not believe that DKO cells are resistant to mitochondrial pore-dependent cell death due to the previously reported mechanism of reduced ER Ca$^{2+}$ levels. Indeed, the initial concept that increased Bcl-2 activity/expression decreases ER Ca$^{2+}$ load by increasing leak, thereby protecting from cell death, is controversial because a number of reports failed to observe any such decrease in ER Ca$^{2+}$ (*Distelhorst and Shore, 2004*).

## Bax/Bak effect on mitochondrial pore-dependent death in conjunction with Bcl-2 proteins

Another issue to address is the potential compensatory role that the prosurvival Bcl-2 family members might play in the absence of Bax/Bak protein in affecting the MPTP-dependent cell death. To examine this issue, we used the Bcl-2/Bcl-xl inhibitor ABT-737 in *Bax/Bak1* DKO and Wt MEFs and examined killing (*Figure 7A*). ABT-737 administration did not restore necrotic killing with ionomycin in *Bax/Bak1* DKO MEFs, although it did increase killing in Wt MEFs (*Figure 7A*). However, gossypol, which can convert endogenous Bcl-2 proteins into a pro-death configuration and increase membrane permeability (*de Peyster et al., 1986*), enhanced or induced ionomycin-dependent cell death in Wt and DKO MEFs, respectively (*Figure 7A*). This result suggests that increasing outer membrane permeability through Bcl-2 family members enhances mitochondrial-dependent necrotic cell death in vivo (*Figure 7A*).

We also analyzed Ca$^{2+}$ uptake capacity in purified mitochondria in the presence of ABT-737 and gossypol to more specifically address the effect on MPTP activity through Bcl-2 family members. ABT-737 did not restore the lack of Ca$^{2+}$ release induced by the MPTP in *Bax/Bak1* DKO mitochondria, while it did hasten MPTP-dependent Ca$^{2+}$ release in Wt mitochondria, collectively suggesting that the prosurvival Bcl-2 family members are not gaining a new protective function in the absence of Bax/Bak protein that might antagonize the MPTP, but that they are working through Bax/Bak (*Figure 7B*). Consistent with the results presented in *Figure 7A*, addition of gossypol to DKO mitochondria restored MPTP function with Ca$^{2+}$ addition, again suggesting that permeabilization of the outer mitochondrial membrane can suffice for Bax/Bak in mediating the final aspects of MPTP-dependent mitochondrial swelling/rupture (*Figure 7B*). We probed this final conclusion in greater mechanistic detail by performing a comparative analysis of Ca$^{2+}$ release from Wt and DKO mitochondria with various agents. After two dosages of Ca$^{2+}$, subsequent gossypol addition induced full Ca$^{2+}$ release in both Wt and DKO mitochondria (*Figure 7C*, top two green colored traces). DKO mitochondria with or without ABT-737 were resistant to Ca$^{2+}$ loss, whereas Wt mitochondria treated with this agent showed partial Ca$^{2+}$ release (*Figure 7C*, compare blue colored traces). This later result shows that by inhibiting the Bcl-2 proteins in Wt mitochondrial, Bax and Bak have even greater activity in the outer mitochondrial membrane leading to enhanced permeability that is more likely to cause swelling with reduced Ca$^{2+}$ release. The fact that DKO mitochondria do not respond whatsoever to ABT-737 demonstrates that the effect is fully dependent on Bax and Bak and their function at the outer membrane (*Figure 7C*).

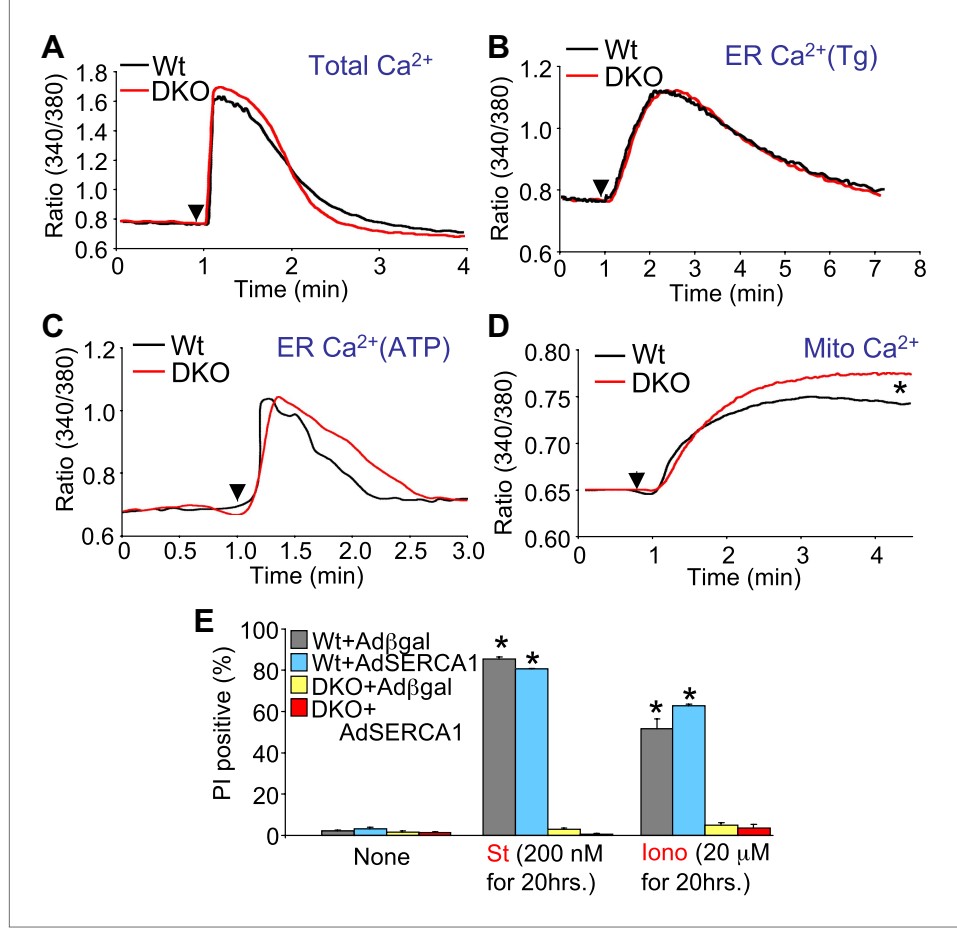

**Figure 6**. Assessment of Ca$^{2+}$ dynamics in *Bax/Bak1*-deficient MEFs. (**A**) Measurement of total cellular Ca$^{2+}$ in Wt and DKO MEFs using the ratiometric Ca$^{2+}$ indicator Fura-2 (read as ratio difference F$_{340}$/F$_{380}$). The arrowhead shows where thapsigargin, FCCP, EDTA, and ionomycin are added to release all intracellular Ca$^{2+}$ from the ER and mitochondria. Two hundred Wt and 320 DKO cells were analyzed. (**B**) and (**C**) Same measurements as in (**A**), except that the SERCA1 inhibitor thapsigargin (**B**) or ATP (**C**) are used to release ER Ca$^{2+}$ over time. The Ca$^{2+}$ signal from DKO MEFs is not significantly different from Wt MEFs. Individual cells were measured on the dish while still alive. One hundred and eighty-five Wt and 282 DKO cells were analyzed for Tg, and 88 Wt and 181 DKO were analyzed for ATP. (**D**) Same measurements as in (**A**) except that only the mitochondrial Ca$^{2+}$ liberating agent FCCP is given. DKO MEFs on a culture dish have greater Ca$^{2+}$ release from their mitochondria than do Wt MEFs (p<0.05), suggesting greater content in the mitochondria at rest. Twenty-six Wt and 44 DKO cells were analyzed. (**E**) Assessment of SERCA1 overexpression in *Bax/Bak1* DKO MEFs and cell death (PI positive). The data show that SERCA1 overexpression in Wt or DKO MEFs does not sensitize to cell death with ionomycin or staurosporine (N = 3 independent experiments).

## Discussion

It is intriguing that loss of Bax/Bak protein dramatically desensitizes the mitochondrial permeability pore, in a manner similar to CsA. Over a decade ago, several reports suggested that Bax and/or Bak directly interacted with presumed components of the MPTP, the adenine nucleotide translocator (ANT) in the inner membrane and the VDAC in the outer membrane (*Marzo et al., 1998*; *Narita et al., 1998*; *Shimizu et al., 1999*). However, while the mitochondrial swelling/permeability pore data reported in these earlier manuscripts are still valid, the conclusion that Bax/Bak function merely to regulate the MPTP by binding VDAC or ANT is not consistent with the data from *Slc25a4/Slc25a5* (ANT1/2 protein) and *Vdac1/2/3* gene–deleted mice, which showed that neither protein is directly required for MPTP formation (*Kokoszka et al., 2004*; *Baines et al., 2007*).

Our data support a model whereby Bax and/or Bak are part of the outer membrane component of the MPTP in a 'resting' nonapoptotic state. Bax has been shown to adopt two types of channel activities,

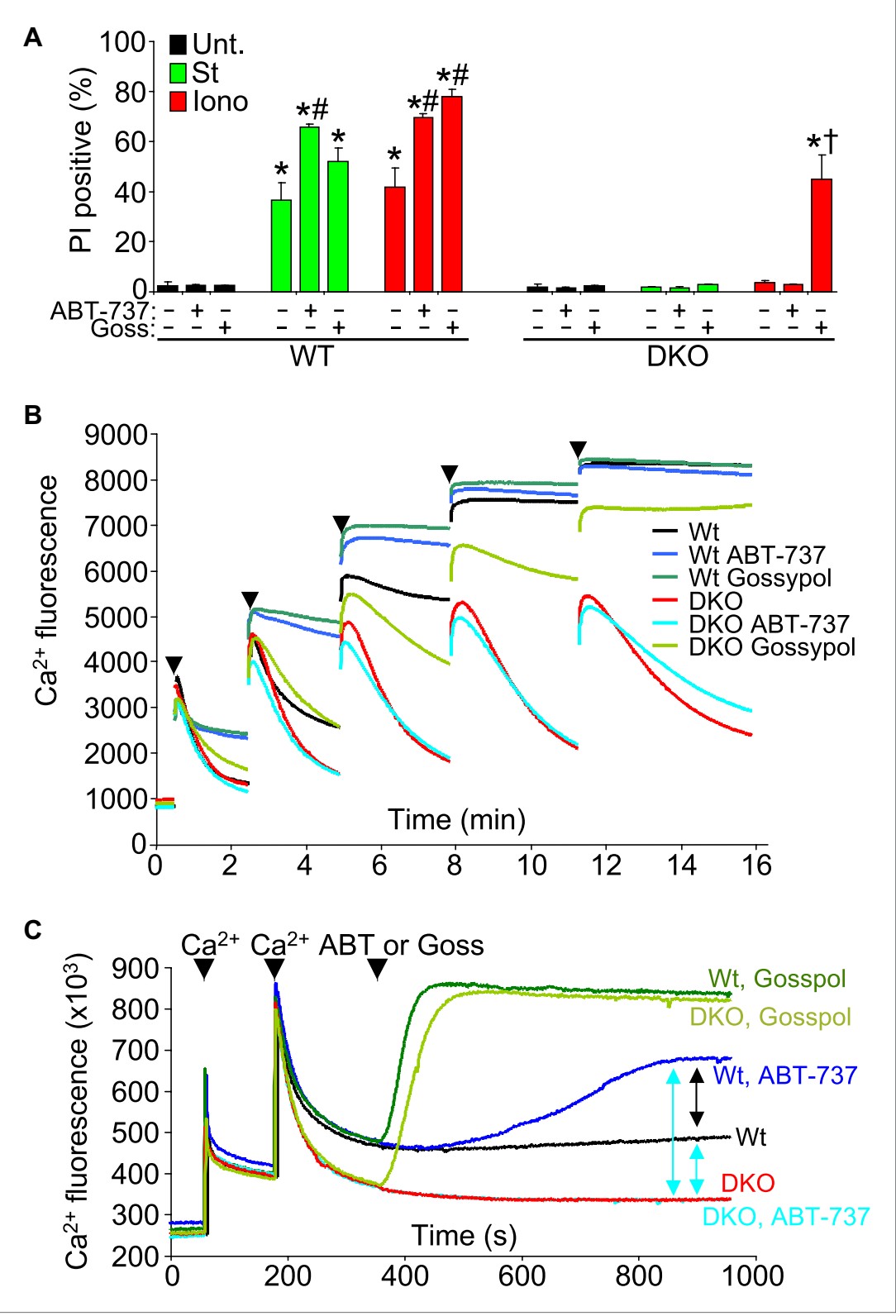

**Figure 7**. Protective Bcl-2 family members are not responsible for the protection observed with *Bax/Bak1* DKO and gossypol restores Ca$^{2+}$-induced killing in DKO cells. (**A**) PI incorporation cell death assay in Wt and DKO MEFs without treatment (none) or with staurosporine (St) or ionomycin (Iono) with or without the Bcl-2/Bcl-xl inhibitor ABT-737 and or gossypol. St was used at 200 nM for 12 hr, ionomycin was used at 20 μM for 24 hr, ABT-737 was
*Figure 7. Continued on next page*

*Figure 7. Continued*

used at 20 µM, and gossypol was used at 10 µM. The results were averaged from three independent experiments. *p<0.05 vs untreated; #p<0.05 vs no ABT-737 or gossypol in St or Iono treated Wt MEFs; †p<0.05 vs ionomycin-treated DKO MEFs. (**B**) $Ca^{2+}$ uptake capacity assay with the external $Ca^{2+}$ indicator dye calcium green-5N and purified mitochondria from Wt or DKO MEFs. 75 µM $Ca^{2+}$ additions are shown at each arrowhead. Fluorescence in the supernatant diminishes as the mitochondria remove the $Ca^{2+}$ from the solution, until the MPTP opens and the $Ca^{2+}$ is no longer sequestered. The swelling and $Ca^{2+}$ uptake assays were performed over three independent experiments, although representative tracings are shown. Gossypol was given as a control for an agent that can increase the permeability of the outer mitochondrial membrane in the absence of Bax/Bak. (**C**) Assay similar to that shown in (**B**) for $Ca^{2+}$ release and MPTP activity under the indicated conditions in purified Wt or DKO mitochondria. The assay was recorded continuously while 50 µM $Ca^{2+}$ was given in two spikes over time, followed by treatment with ABT-737 or gossypol (given at the arrowheads).

one of which has a small pore size (0.9 nm) and is homogenous in the outer mitochondrial membrane, which could fulfill the permeability function we observed here to permit nonapoptotic cell death (*Lin et al., 2011*). As another consideration, the α5/α6 domain of Bax is known to have membrane permeation activity and to permit pore formation on its own, the permeation size of which appears to reach an equilibrium of less than 10 kDa (*Fuertes et al., 2010*). Bcl-2 family members appear to generally alter the permeability and rigidity of lipid membranes, and even the Bcl-2 homologue from *Caenorhabditis elegans*, CED-9 can induce leakiness of reconstituted lipid membranes (*Tan et al., 2011*). The exact mechanism whereby Bcl-2 family members might increase membrane permeability and small molecular permeation in their nonoligomerized state is unknown, although it may be related to the distinct lipid environment in the mitochondria and how it interacts with the select hydrophobic α-helices in the core of the Bcl-2 proteins.

The most straightforward model that is consistent with our data and the literature describing the biophysical properties of Bcl-2 family members is that Bax and Bak function as the outer membrane activity of the MPTP. Once the inner membrane pore opens through a CypD-regulated event, the outer membrane is already poised to complete the process, given the permeability characteristics imparted by Bax/Bak in their monomeric states (*Figure 8*). Indeed, liposome reconstitution with recombinant nonactivated Bax directly increased membrane permeability, and rBax immediately restored swelling in isolated mitochondria. Moreover, increasing the activity of Bax/Bak with ABT-737 sensitized the MPTP to opening with mild $Ca^{2+}$ stimulation in Wt but not DKO mitochondria, and nonspecific increases in outer mitochondrial membrane permeability with gossypol or other agents fully restored mitochondrial swelling and cell death in the absence of Bax/Bak. Direct patching of mitochondria outer vs inner membranes showed that Bax/Bak permit MPTP-dependent mitochondrial swelling by only enhancing conductance of the outer membrane. Our results also suggest that the outer mitochondrial membrane does not directly induce the inner membrane to undergo MPTP, and that initiation of this process is an inner membrane-regulated phenomenon (*Figure 8*). However, lack of sufficient outer membrane permeability, as observed in the absence of Bax/Bak protein or with addition of poloxamer-188 in Wt mitochondria, sufficiently restrains swelling and organelle rupture to allow reestablishment of inner membrane potential and continued cell survival. Thus, Bax and Bak are necessary for MPTP-dependent cell death by functioning exclusively within the outer membrane as permeability factors that facilitate an irreversible swelling threshold leading to rupture of the mitochondrion and cell death, once the inner membrane component is fully engaged.

While the role of Bax/Bak might appear to be passive in permitting mitochondrial swelling and MPTP-dependent death by augmenting the permeability characteristics of the outer membrane, the entire process is clearly not passive and should still be subject to acute regulation through other Bcl-2 family members that affect Bax/Bak. Indeed, ABT-737 treatment increased cellular necrosis, mitochondrial swelling, and enhanced $Ca^{2+}$ release in isolated mitochondria by augmenting the amount/activity of Bax/Bak in the outer mitochondrial membrane. While we cannot rule out other potential mechanisms of action for ABT-737, our results are consistent with the hypothesis that acute alterations in the activity of select Bcl-2 family members could affect cellular necrosis by acutely regulating the activity, localization, or function of Bax/Bak. Indeed, Nix/BNip3L, which is induced by disease/developmental signaling pathways that enhance cell death, can alter the activity and pro-death characteristics of Bax/Bak as well as alter MPTP-dependent cell death (*Chen et al., 2010*).

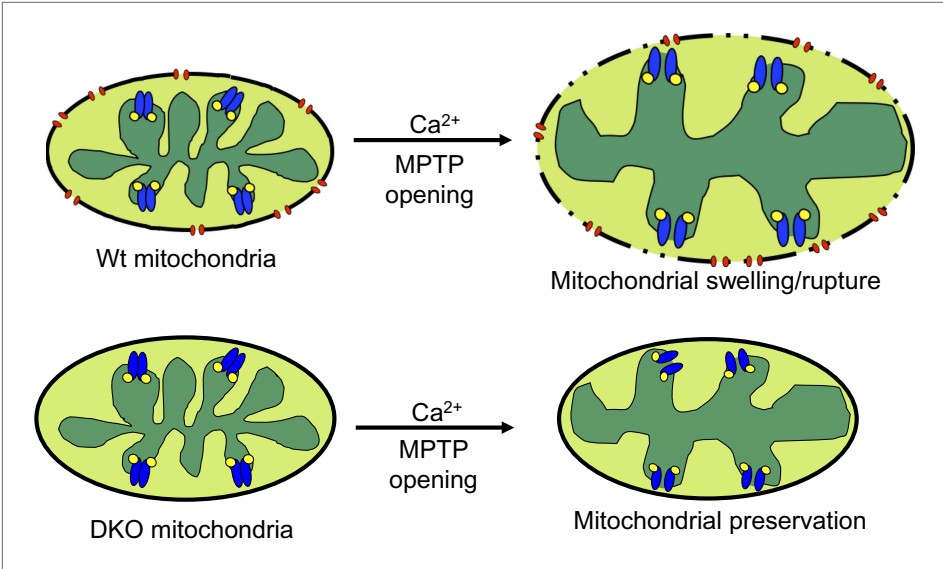

**Figure 8**. Schematic representation of how Bax and Bak influence MPTP-dependent mitochondrial swelling and organelle rupture. The model shows mitochondria undergoing MPTP opening (blue) via CypD (yellow) in the presence and absence of Bax/Bak (red). When Bax/Bak are present on the outer membrane and the MPTP opens in response to $Ca^{2+}$, it causes the mitochondrial inner membrane to dissipate its electrochemical gradient leading to additional swelling and eventually rupture of the outer membrane and entire organelle. When Bax/Bak are absent, the outer membrane has lower permeability, which prevents swelling and rupture and subsequent necrosis even though the inner membrane has undergone MPTP opening.

With respect to medical relevance, our results are important because they suggest that Bax/Bak are a common nodal point in both apoptotic and necrotic cell death. Thus, inhibition of Bax/Bak should be the most therapeutically potent means of antagonizing acute cell death following ischemic injury in vivo or in adult onset degenerative diseases, as it would block both mitochondrial-dependent processes. Moreover, other types of regulated necrotic cell death might also be inhibited in the absence of Bax/Bak, such as necroptosis, also referred to as extrinsic necrosis or RIP-dependent necrosis (*Degterev et al., 2005*; *Hitomi et al., 2008*; *Wang et al., 2012*). Indeed, we and others have observed that *Bax/Bak1* null MEFs were resistant to this form of cell death (data not shown and *Irrinki et al., 2011*). Necroptosis is induced in cells when stimulated with tumor necrosis factor α (TNFα) in the presence of caspase inhibitors, such as zVAD-fmk (*Degterev et al., 2005*). Necroptosis is inhibited with by necrostatins, a series of serine–threonine kinase inhibitors that were later shown to block RIP1 kinase in preventing TNFα- and ZVAD-dependent necroptosis in culture. RIP1 is able to interact with RIP3 in controlling necroptosis (*Cho et al., 2009*; *He et al., 2009*; *Zhang et al., 2009*). Recently, mixed lineage kinase domain–like protein (MLKL) and phosphoglycerate mutase family member 5 (PGAM5) were shown to be downstream of the RIP proteins (*Sun et al., 2012*; *Wang et al., 2012*). Knockdown of either MLKL or PGAM5 resulted in protection against RIP-mediated necroptosis, as well as protection against ROS or $Ca^{2+}$ overload–mediated necrosis (*Wang et al., 2012*). This duality led the authors to recognize the existence of two arms of the necrotic pathway, the extrinsic arm (TNF + zVAD–induced necrosis) and the intrinsic arm that responds to ROS or $Ca^{2+}$ overload–induced necrosis, which is the form of necrosis that we investigated here as affected by Bax/Bak (*Wang et al., 2012*).

## Materials and methods

### Tissue culture and analysis of cell death and viability

Wt and *Bax* and *Bak1* DKO SV40 immortalized MEFs, and all variations were cultured in IMDM medium supplemented with 10% bovine growth serum, antibiotics, and nonessential amino acids. DKO MEFs expressing Wt and mutant Bax-GFP were previously described (*Wei et al., 2001*; *Hoppins et al., 2011*). DKO MEFs with reconstituted Bax and Bak were also previously described (*Kim et al., 2009*).

DKO MEFs were infected with a binary adenovirus expressing Bax (Ad-Bax) was previously described and obtained from Dr Bingliang Fang (*Kagawa et al., 2000*). DKO MEFs were also infected with an adenovirus expressing β-galactosidase (Adβgal, control) or sarco/endoplasmic reticulum $Ca^{2+}$-ATPase (AdSerca1) (Vector BioLabs, Philadelphia, PA). *Ppif* null MEFs were described previously (*Baines et al., 2005*), although for the current experiments, the cells were subjected to SV40-mediated immortalization. At 80% confluence, MEFs were treated with 200 nM staurosporine for 2–24 hr, 20 µM ionomycin for 4–24 hr, 750–800 µM $H_2O_2$ for 2–24 hr, 750 µM MMS for 2–24 hr, or 60–90 ng TNFα and 40 µM caspase inhibitor z-Vad-FMK (z-Vad; Promega, Madison, WI) for 24 hr (in some experiments, cells were pretreated with 40 µM z-Vad, 20 µM ABT-737, or 10 µM Gossypol). Cell death and viability was determined by PI uptake and annexin V (AV) positivity (BioVision, Milpitas, CA). Briefly, cells were trypsinized and washed twice and incubated with AV and PI for 10 min. The cells were then quantified for PI and AV positivity at 10,000 counts per sample by using a Cell Lab Quanta SC flow cytometer (Beckman Coulter, Indianapolis, IN). In some experiments, only PI was used and AV was replaced with PBS. Cell death and viability were also measured using the MultiTox-Fluor Multiplex Cytotoxicity Assay (Promega). Briefly, cells were cultured in 96-well plates and treated for given times and dosages, after which they were incubated with 2× MultiTox-Fluor Multiplex Cytotoxicity Assay reagent for 30 min at 37°C. The assay quantifies dead/live ratios by measuring membrane rupturing or opening, such as during necrosis or apoptosis, which allows access of proteases to the fluorogenic peptide substrate that was quantified by using Synergy 2 Multi-Mode Microplate Reader (BioTek, Winooski, VT). Oxygen consumption was measured with an XF extracellular flux analyzer (Seahorse Bioscience, North Billerica, MA). Cells were plated to 100% confluence on XF24 cell culture microplates. After an initial measurement, the wells were injected with St, Iono, or vehicle, and oxygen consumption was measured every 6 hr for 24 hr.

## Mitochondrial isolation and analyses

Liver, heart, and MEF mitochondria were isolated by homogenization followed by differential centrifugation. Livers and hearts were prepared with a Teflon homogenizer, while cells were disrupted with a glass homogenizer. The isolation buffer consisted of 250 mM sucrose and 10 mM Tris pH 7.4. Mitoplasts were isolated by incubating the isolated mitochondria in a hypotonic buffer, which consisted of 10 mM KCl, 10 mM Tris pH 7.4 for 20 min followed by gentle agitation with a pipette, and a low-speed centrifugation.

Mitochondrial swelling was performed on either 0.5 or 1 mg of mitochondria in 1 ml with light scattering measured at 540 nm. The mitochondrial swelling buffer consisted of 120 mM KCl, 10 mM Tris pH 7.4, 5 mM $KH_2PO_4$, 7 mM pyruvate, 1 mM malate, and 10 µM EDTA. Swelling was induced by either 400 or 800 µM $CaCl_2$ or 200 µM atractyloside in EDTA-free buffer. In some experiments, mitochondria were also treated with 10 µM gossypol, 9 nM tetanolysin, 25 µM digitonin, 1–10% poloxamer-188, 2 µM CsA, or 5 µg of monomeric recombinant Bax-GST (rBax). Mitoplast swelling was performed identically to mitochondrial swelling except that the swelling buffer consisted of 10 mM KCl. We determined that the rBax was not causing cytochrome *c* release by treating mitochondria from *Bax-loxP (fl) Bak1*$^{-/-}$ Albumin-Cre livers with 5 µg rBax alone or with 2.5 µg recombinant tBid and 5 µg rBax as a positive control or vehicle. The mitochondria were incubated in swelling buffer containing 100 mM KCl for 10 min at room temperature to emulate the swelling assay. After incubation, the mitochondria were centrifuged and the supernatant was subjected to Western blot analysis for cytochrome *c*.

Cytochrome *c* release was detected from whole cells treated with 200 nM St, 800 µM $H_2O_2$, or 20 µM Iono for 6 hr by Western blot analysis. Cytosolic fractions were obtained by homogenizing the treated cells in mitochondrial isolation buffer containing 100 mM KCl. Homogenates were then centrifuged at 14,000×*g* for 10 min; the cytosolic fractions were collected and subjected to an additional spin and then Western blotting.

Mitochondrial $Ca^{2+}$ uptake was measured with Calcium Green-5N (Invitrogen, Grand Island, NY) as previously described (*Kwong et al., 2007*). Briefly, MEF mitochondria were isolated in MS-EGTA buffer (225 mM mannitol, 75 mM sucrose, 5 mM HEPES, and 1 mM EGTA, pH 7.4). Mitochondria (150 or 200 µg for 96-well format or 500 µg for single cuvette format) were then incubated in KCl buffer (125 mM KCl, 20 mM HEPES, 2 mM $MgCl_2$, 2 mM $KH_2PO_4$, and 40 µM EGTA, pH 7.2) containing 200 nM Calcium Green-5N, 7 mM pyruvate, and 1 mM malate. Mitochondria were treated with sequential additions of $CaCl_2$ (50–100 µM). Fluorescence was quantified using a Synergy 2 Multi-Mode Microplate Reader (BioTek) or by cuvette-based fluorometric analysis using a fluorometer (Photon Technology International [PTI], Birmingham, NJ). For some experiments, mitochondria were treated with 10 µM gossypol, 20 µM ABT-737, or 1–10% poloxamer-188.

Inner mitochondrial membrane permeability was assessed by using the calcein/cobalt assay. Briefly, cells were incubated with 1 µM Calcein-AM and 8 mM $CoCl_2$ for 10 min and then treated with 200 nM St or 20 µM Iono for 15 min. Cells were imaged by confocal microscopy. Mitochondrial oxygen consumption was measured with an XF extracellular flux analyzer. Mitochondria (150 µg) were incubated in KCl buffer containing 7 mM pyruvate and 1 mM malate. Mitochondria were treated with four sequential additions of $CaCl_2$ (200 µM). Oxygen consumption was measured 5 min after every injection.

## Animal models

*Bax-loxP (fl) Bak1*$^{-/-}$ mice were described previously (*Takeuchi et al., 2005*). To create heart- and liver-specific *Bax* and *Bak1* knockout mice, *Baxfl/fl Bak1*$^{-/-}$ mice were crossed with α-myosin heavy chain (αMHC)-Cre and albumin-Cre transgenic lines (*Postic et al., 1999*; *Oka et al., 2006*). IR injury was performed as previously described (*Kaiser et al., 2004*). Mice were also subjected to permanent ligation of the left anterior descending coronary artery to induce myocardial infarctions. These mice were monitored daily for 8 weeks, and death events were recorded.

## Histological analyses

Electron microscopy was performed on MEFs and mitochondria isolated from Wt Albumin-Cre and *Baxfl/fl Bak1*$^{-/-}$ Albumin-Cre mouse livers. Prior to fixation, the mitochondria were subjected to the mitochondrial swelling assay. Samples were then fixed in glutaraldehyde and cacodylate, embedded in epoxy resin, and sectioned. Sections were counterstained with uranyl acetate and lead citrate. Mitochondrial cross-sectional area was quantified by using image-J software (NIH). Histological analysis of the injury induced by myocardial ischemia/reperfusion was described previously (*Kaiser et al., 2004*).

## Western blotting and immunoprecipitation

Livers, hearts, and MEFs were homogenized in RIPA buffer containing protease inhibitor cocktails (Roche, Indianapolis, IN). The following antibodies were used: Bax (Santa Cruz, Santa Cruz, CA), active Bax (Exalpha Biologicals, Shirley, MA), Bak (Millipore, Billerica, MA), Bcl-2 (Santa Cruz), Bcl-xl (Santa Cruz), Bid (Santa Cruz), Bim (BD Biosciences, San Jose, CA), BNip3 (Cell Signaling, Danvers, MA), VDAC 1 (MitoSciences, Eugene, OR), ANT 1 (EMD4Biosciences, Billerica, MA), CypD (MitoSciences), cytochrome *c* (BD biosciences), complex I-V (MitoSciences), Sam50 (Sigma, St. Louis, MO), and β-tubulin (Santa Cruz). Mitochondria from Wt and DKO MEFs were lysed in IP buffer (20 mM Tris pH 7.4, 150 mM NaCl, 1% triton X-100, 0.3 mM PMSF, 0.5 mM DTT, and 1× phosphatase inhibitors). Lysates were incubated with primary antibody for Bax and protein A/G plus agarose (Santa Cruz) beads for 12 hr at 4°C. The lysates were centrifuged and washed three times and boiled in SDS sample buffer.

## Bax/Bak activation and gel-filtration analyses

Flow cytometry analysis of active Bax has been previously described (*Panaretakis et al., 2002*). Briefly, cells were harvested and fixed in 0.25% paraformaldehyde for 5 min, washed three times in PBS, and incubated with primary antibody diluted at 1:50 in PBS-containing digitonin (100 µg/ml) for 30 min. The cells were then washed three times in PBS and incubated with FITC-labeled secondary antibody for 30 min, washed, and resuspended in PBS and analyzed with a flow cytometer (10,000/sample). Activation of Bak was determined by cross-linking isolated mitochondria (50 µg) from cells treated with 10 mM disuccinimidyl suberate (DSS) at room temperature for 30 min. The mitochondria were then suspended in SDS sample buffer, and a Western blot was performed. Wt MEFs were treated with or without St and Iono for 12 hr in the presence of 40 µM z-Vad. After 12 hr, they were washed with ice-cold PBS and lysed in HNC buffer (25 mM HEPES, pH 7.5, 300 mM NaCl, 1 mM DTT, and 2% CHAPS). The lysates were loaded onto a Superdex 200 HR 10/30 column (GE, Pittsburgh, PA). The column was pre-equilibrated with a buffer consisting of 25 mM HEPES, pH 7.5, 300 mM NaCl, 0.2 mM DTT, and 2% CHAPS and calibrated with ferritin (440 kDa), B amylase (200 kDa), alcohol dehydrogenase (150 kDa), BSA (66 kDa), carbonic anhydrase (29 kDa), and cytochrome *c* (12 kDa; Sigma). The proteins were collected in 0.6 ml fractions every minute. The fractions were then subjected to TCA precipitation to concentrate the protein before Western blotting.

## Cytosolic Ca²⁺ measurements

MEFs were plated on glass bottom dishes (MatTek, Ashland, MA). Cells were loaded in filtered Ringer's solution consisting of 145 mM NaCl, 5 mM KCl, 2 mM $CaCl_2$, 1 mM $MgCl_2$, and 10 mM HEPES with 5 µM Fura-2 AM for 30 min at room temperature, washed twice, and incubated for an additional 30 min before beginning the experiment. Experiments were performed on a Nikon Eclipse Ti-U inverted microscope

equipped with a Delta Scan dual-beam spectrofluorophotometer (PTI) set to excite at 340 and 380 nm. A Photometrics CoolSnap ES$^2$ camera was used to acquire images at 510 nm. EasyRatioPro software was used to gather and analyze the data. Experiments were performed by imaging cells at 1 Hz for 2 min in calcium- and magnesium-free (CMF) solution to obtain a basal calcium reading. After 2 min, the cells were treated with the following: 1 µM thapsigargin (Tg), 100 µM ATP, 2 µM FCCP, or a combination of 5 µM Iono, 1 µM Tg, 2 µM FCCP, and EDTA. Imaging continued at 1 Hz for 5 min.

## Atomic mass spectrometry

Mitochondria from Wt and DKO MEFs were subjected to the mitochondrial swelling assay for 5 min. They were then centrifuged and were analyzed for $^{39}$K, $^{43}$Ca, $^{23}$Na, $^{31}$P, and $^{24}$Mg. Each sample was analyzed with a 100× dilution. Typical standards ranging from 0 to 5000 ppb were used for the calibration curves, and most curves were determined to be $R^2$ = 0.998–0.999. Internal standards $^9$Be and $^{89}$Y were used during analysis. An Agilent ICPMS 7700× (Agilent Technologies, Santa Clara, CA) was used for the element-specific detection. The ICPMS was equipped with a microconcentric nebulizer supplied by Glass Expansion (Pocasset, MA), a Scott-type double-channel spray chamber (cooled to 2°C), a shield torch, an octopole collision/reaction cell with pressurized helium gas (purity of 99.999%), a quadrupole mass analyzer, and an electron multiplier. Instrumental parameters: RF forward power = 1500 W, plasma Ar gas flow rate = 15.0 l/min, carrier Ar gas flow rate = 1.07 l/min, and He collision gas = 4.5 ml/min. Resulting data were analyzed using Agilent Mass Hunter ICPMS software.

## Patch clamping techniques

Cells at 80% confluence were harvested and mitochondria were isolated as previously described (*Murphy et al., 2001*). Membrane patches of isolated mitochondria were excised after formation of a seal using micropipettes with approximately 0.3-µm tips and resistances of 10–30 MΩ at room temperature. Patching media was 150 mM KCl, 5 mM HEPES, 0.23 mM CaCl$_2$, 1 mM EGTA, pH 7.4. Voltages were clamped with an Axopatch 200 amplifier and reported as pipette potentials. Permeability was typically determined from stable current levels and/or total amplitude histograms of 30 s of data at +20 mV. pClamp version 8 (Axon Instruments, Sunnyvale, CA) and WinEDR v2.3.3 (Strathclyde Electrophysiological Software, Glasgow, UK) were used for current analysis as previously described (*Martinez-Caballero et al., 2009*). Sample rate was 5 kHz with 1–2 kHz filtration. The channel activity of rBax was characterized in DKO mitochondria and liposomes devoid of other proteins with and without its activator tBid. Micropipette tips were filled with media containing 10–100 ng/µl monomeric rBax and then backfilled with patching media. Hence, the actual Bax concentration was lower than that loaded in the tips. Seals were formed with these micropipettes on giant liposomes or mitochondria prepared as described previously (*Martinez-Caballero et al., 2009*). Ion selectivity was determined through reversal potentials after a 150:30 mM KCl gradient was established across the patch as previously reported (*Pavlov et al., 2001*).

## Acknowledgements

We thank Suzanne Hoppins and Jodi Nunnari (University of California, Davis, CA) for providing the MEFs with the non-oligomerizing Bax mutants. JK was supported by a local Affiliate American Heart Association pre-doctoral grant.

## Additional information

### Funding

| Funder | Grant reference number | Author |
|---|---|---|
| Howard Hughes Medical Institute | | Jeffery D Molkentin |
| National Institutes of Health | P01 HL080101 | Jeffrey Robbins, Kathleen W Kinnally, Jeffery D Molkentin |
| American Heart Association pre-doctoral grant | | Jason Karch |

The funders had no role in study design, data collection and interpretation, or the decision to submit the work for publication.

## Author contributions

JK, Conception and design, Acquisition of data, Analysis and interpretation of data, Drafting or revising the article; JQK, Acquisition of data, Analysis and interpretation of data; ARB, MAS, PMP, SM-C, Acquisition of data; JWE, Conception and design, Contributed unpublished essential data or reagents; HO, JR, Provided the TEM data; EH-YC, Provided critical cell lines lacking *Bax/Bak1* that were reconstituted with mutant forms of Bax that are non-apoptotic; KWK, Conception and design, Acquisition of data, Analysis and interpretation of data; JDM, Conception and design, Analysis and interpretation of data, Drafting or revising the article

## Ethics

Animal experimentation: All experiments involving animals were approved by the IACUC under protocols 2D01003 and 2E11104 to Dr Molkentin.

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
