## [Decision Letter]

Thank you for sending your work entitled “Bax and Bak function as the outer membrane component of the mitochondrial pore in regulating cell death” for consideration at *eLife*. Your article has been evaluated by a Senior editor and 2 reviewers.

The Senior editor and the two reviewers discussed their comments before we reached this decision, and the Senior editor has assembled the following comments to help you prepare a revised submission.

This manuscript addresses the important question of whether Bax and Bak regulate the permeability properties of the outer mitochondrial membrane. The data suggest that Bax/Bak null mitochondria resist swelling, and they are interpreted within a model “whereby Bax and/or Bak are part of the outer membrane component of the MPTP in a ‘resting’ non-apoptotic state”. Yet the mitochondrial permeability transition pore (MPTP) is an inner membrane event that readily occurs in mitoplasts. The link with MPTP-dependent swelling is therefore an interpretation that must be confirmed by additional experiments. The proposed experiments will address whether Bax/Bak affect outer membrane resistance to rupture only when the MPTP opens or independent of the mechanism of matrix swelling.

1) The first experiment the reviewers agreed on is to test whether mitochondria are able to resist swelling if the mitoplast swelling is induced by other agents that are not known to cause MPTP, for example valinomycin in K^+^- and Pi-containing media, or in iso-osmotic media containing varying K^+^ concentrations, best if in the presence of added cytochrome c. This would also allow testing to determine whether resistance to swelling is specifically seen when the pore opens, or is rather an independent feature conferred to the outer membrane by lack of Bax/Bak.

2) Additionally, since comparing mitochondria of different genotypes is not trivial, and normalization to initial absorbance can be deceptive, what is needed is the addition of a pore-forming agent (such as alamethicin) at the end of each swelling experiment, which should report the actual rather than normalized absorbance values to test whether the Bax/Bak null mitochondria can swell, and to what extent. If the Bax/Bak null mitochondria are resistant to alamethicin this would already indicate that lack of swelling is not specifically related to the permeability transition; if they do swell this would make for an important internal calibration allowing a comparison with the wild type genotype.

Although we agree that the main observation is of interest, the interpretation of such may change depending on the outcome of suggested experiments.

---

## [Author Response]

*1) The first experiment the reviewers agreed on is to test whether mitochondria resistant to swelling can happen if the mitoplast swelling is induced by on other agents that not known to cause MPTP*.

We agree: determining whether lack of Bax/Bak specifically affects MPTP dependent swelling or all types of mitochondrial swelling is crucial to this manuscript. To address this issue we isolated both wild type and Bax/Bak deficient mitochondria from MEFs and livers. These mitochondria were subjected to varying concentrations of KCl. Wild type and Bax/Bak null mitochondria responded equally to the varying concentrations of KCl as now seen in the newly added Figure 2—figure supplement 4 (MEFs) & G,H (livers). These new data further strengthen our hypothesis that Bax and Bak are required specifically for MPTP dependent mitochondrial swelling and not non-specific mitochondrial swelling induced by osmotic changes.

*2) Additionally, since comparing mitochondria of different genotypes is not trivial, and normalization to initial absorbance can be deceptive, what is needed is the addition of a pore-forming agent (such as alamethicin) at the end of each swelling experiment, which should report the actual rather than normalized absorbance values to test whether the Bax/Bak null mitochondria can swell, and to what extent*.

This is a good suggestion and we have added such data. As requested we now show raw absorbance that is not normalized. However, in the past we normalized such values because the initial absorbance can slightly vary from preparation to preparation, although independent of genotype (it is a variable of such purification preparations). Hence, while initial absorbance varies slightly between preparations, the response of the mitochondria in each preparation is the important value and this is always consistent, similar to normalized data. In all of our swelling experiments we have never observed a significant difference in the starting point between any of our genotypes (hence the subtle variation in starting swollen state is an independent variable that does not track with any genotype).

Indeed, the new data we are showing in Figure 2—figure supplement 4 show both raw (A, E) and normalized (D, F) values, which show the same relative changes after calcium and then alamethicin. These data were collected from both purified mitochondria isolated from MEFs and livers that are control or Bax/Bak deficient. In both mitochondria, calcium induced swelling does not occur when Bax and Bak are absent, but the addition of alamethicin confirms that these mitochondria have an equal maximal swelling potential when compared to the wild type control. Taken together, the results from the newly added Figure 2—figure supplement 4 reinforces our original hypothesis.